# Frequent Germline and Somatic Single Nucleotide Variants in the Promoter Region of the Ribosomal RNA Gene in Japanese Lung Adenocarcinoma Patients

**DOI:** 10.3390/cells9112409

**Published:** 2020-11-03

**Authors:** Riuko Ohashi, Hajime Umezu, Ayako Sato, Tatsuya Abé, Shuhei Kondo, Kenji Daigo, Seijiro Sato, Norikazu Hara, Akinori Miyashita, Takeshi Ikeuchi, Teiichi Motoyama, Masashi Kishi, Tadahiro Nagaoka, Keiko Horiuchi, Atsushi Shiga, Shujiro Okuda, Tomoki Sekiya, Aya Ohtsubo, Kosuke Ichikawa, Hiroshi Kagamu, Toshiaki Kikuchi, Satoshi Watanabe, Jun-Ichi Tanuma, Peter Schraml, Takao Hamakubo, Masanori Tsuchida, Yoichi Ajioka

**Affiliations:** 1Histopathology Core Facility, Niigata University Faculty of Medicine, Niigata 951-8510, Japan; ajioka@med.niigata-u.ac.jp; 2Division of Molecular and Diagnostic Pathology, Niigata University Graduate School of Medical and Dental Sciences, Niigata 951-8510, Japan; pathchem@med.niigata-u.ac.jp (A.S.); abet@med.niigata-u.ac.jp (T.A.); koptaro_21@yahoo.co.jp (S.K.); tmotoyama78@email.plala.or.jp (T.M.); 3Division of Pathology, Niigata University Medical & Dental Hospital, Niigata 951-8520, Japan; umezu@med.niigata-u.ac.jp; 4Division of Oral Pathology, Department of Tissue Regeneration and Reconstruction, Niigata University Graduate School of Medical and Dental Sciences, Niigata 951-8514, Japan; tanuma@dent.niigata-u.ac.jp; 5Department of Protein-Protein Interaction Research, Institute for Advanced Medical Sciences, Nippon Medical School, Kawasaki, Kanagawa 211-8533, Japan; kenji-daigo@nms.ac.jp (K.D.); keiko-horiuchi@nms.ac.jp (K.H.); t-hamakubo@nms.ac.jp (T.H.); 6Division of Thoracic and Cardiovascular Surgery, Niigata University Graduate School of Medical and Dental Sciences, Niigata 951-8510, Japan; seisato@med.niigata-u.ac.jp (S.S.); masatsu@med.niigata-u.ac.jp (M.T.); 7Department of Molecular Genetics, Brain Research Institute, Niigata University, Niigata 951-8510, Japan; nhara@bri.niigata-u.ac.jp (N.H.); miyashi2020@bri.niigata-u.ac.jp (A.M.); ikeuchi@bri.niigata-u.ac.jp (T.I.); 8Neuroscience Laboratory, Research Institute, Nozaki Tokushukai Hospital, Osaka 574-0074, Japan; masashi.kishi@gmail.com; 9Division for Therapies against Intractable Diseases, Institute for Comprehensive Medical Science, Fujita Health University, Aichi 470-1192, Japan; nagaokat@fujita-hu.ac.jp; 10Department of Clinical Laboratory, Niigata Cancer Center hospital, Niigata 951-8560, Japan; shiga0410@gmail.com; 11Division of Bioinformatics, Graduate School of Medical and Dental Sciences, Niigata University, Niigata 951-8510, Japan; okd@med.niigata-u.ac.jp; 12Department of Respiratory Medicine and Infectious Diseases, Niigata University Graduate School of Medical and Dental Sciences, Niigata 951-8510, Japan; sekiya.trueseal.9@gmail.com (T.S.); ayaohtsubo@yahoo.co.jp (A.O.); wrx5772@yahoo.co.jp (K.I.); kagamu19@saitama-med.ac.jp (H.K.); kikuchi@med.niigata-u.ac.jp (T.K.); satoshi7@med.niigata-u.ac.jp (S.W.); 13Present address: Division of Respiratory Medicine, Saitama Medical University International Medical Center, Saitama 350-1298, Japan; 14Department of Pathology and Molecular Pathology, University of Zurich and University Hospital Zurich, CH-8091 Zurich, Switzerland; Peter.Schraml@usz.ch

**Keywords:** lung adenocarcinoma, rDNA, ribosome biogenesis, single nucleotide variant, prognosis

## Abstract

Ribosomal RNA (rRNA), the most abundant non-coding RNA species, is a major component of the ribosome. Impaired ribosome biogenesis causes the dysfunction of protein synthesis and diseases called “ribosomopathies,” including genetic disorders with cancer risk. However, the potential role of rRNA gene (rDNA) alterations in cancer is unknown. We investigated germline and somatic single-nucleotide variants (SNVs) in the rDNA promoter region (positions −248 to +100, relative to the transcription start site) in 82 lung adenocarcinomas (LUAC). Twenty-nine tumors (35.4%) carried germline SNVs, and eight tumors (9.8%) harbored somatic SNVs. Interestingly, the presence of germline SNVs between positions +1 and +100 (*n* = 12; 14.6%) was associated with significantly shorter recurrence-free survival (RFS) and overall survival (OS) by univariate analysis (*p* < 0.05, respectively), and was an independent prognostic factor for RFS and OS by multivariate analysis. LUAC cell line PC9, carrying rDNA promoter SNV at position +49, showed significantly higher ribosome biogenesis than H1650 cells without SNV. Upon nucleolar stress induced by actinomycin D, PC9 retained significantly higher ribosome biogenesis than H1650. These results highlight the possible functional role of SNVs at specific sites of the rDNA promoter region in ribosome biogenesis, the progression of LUAC, and their potential prognostic value.

## 1. Introduction

Lung cancer is the leading cause of cancer-related deaths worldwide, and non-small cell lung cancer (NSCLC) accounts for most cases (approximately 85%) [1]. Lung adenocarcinoma (LUAC) is the most common histological subtype of NSCLC [1,2,3]. Numerous studies have identified biologically significant genetic alterations involved in various aspects of NSCLC, particularly of LUAC, including carcinogenesis, cancer progression, prognosis and treatment. Recent comprehensive studies showed that mutations in *TP53*, *STK11*, or *SMARCA4* are associated with poor prognosis [4,5]. The identification of several types of genetic alterations, such as mutations in *EGFR* or *BRAF*, and gene fusion events involving *ALK*, *ROS1* and *RET*, provided potential therapeutic targets [1,4,5,6]. Besides protein-coding mRNA, there are different types of non-coding RNA (ncRNA) species comprising RNA molecules that are not translated into proteins. While most cancer genome studies focused exclusively on variants that alter the amino acid sequences of protein-coding genes [7], in recent years, a new and crucial role of ncRNA in cancer has been highlighted. Although initially ncRNAs were thought to be “junk”, with no functional purpose, trailblazing research has discovered their diverse functions in coding, decoding, transfer, regulation, gene expression, and complex integrated networks [7,8,9,10]. Now the huge contribution of ncRNAs in cancer development and progression has been recognized. Two of the most extensively studied ncRNA classes, long non-coding RNAs (lncRNAs) and microRNAs (miRNAs), particularly play critical roles in human cancer. Numerous large-scale studies have shown that many lncRNAs and miRNAs may represent possible promising diagnostic and prognostic biomarkers [10].

The ribosomal RNA gene (rDNA) encodes ribosomal RNA (rRNA), which is the most abundant ncRNA, accounting for ~90% of the total RNA present in a cell [8,9]. The main rDNA, i.e., 45S rDNA, comprises several hundred transcription units organized in tandem repeats and clustered on five chromosomal loci on chromosomes 13, 14, 15, 2, and 22 in humans [11]; the transcription units are transcribed solely by RNA polymerase I (Pol I) in the nucleolus to produce the 47S/45S precursor rRNA (pre-rRNA). This pre-rRNA transcript is then processed into three mature rRNAs (18S, 5.8S, and 28S rRNAs). The last small molecule, 5S rRNA, is synthesized outside the nucleolus by RNA polymerase III (Pol III) [12,13,14,15,16]. Concurrent with transcription and ribosomal protein–pre-rRNA assembly, pre-rRNAs are extensively processed and modified, ultimately forming ribosomes, which are cellular organelles responsible for the machinery of protein synthesis. These coordinated processes, from the initial rDNA transcription to the creation of mature ribosomes, are called ribosome biogenesis [16,17]. Impaired ribosome biogenesis causes a series of diseases known as “ribosomopathies,” including genetic disorders such as Diamond-Blackfan anemia, Shwachman–Diamond syndrome, X-linked dyskeratosis congenita and cartilage hair hypoplasia, which stem from impaired protein synthesis resulting from dysfunctional ribosomes. Some data suggest an active role for ribosome biogenesis in carcinogenesis through its effect in balancing protein translation, thereby altering the synthesis of proteins that play an important role in the genesis of cancer and resulting in uncontrolled cancer cell proliferation [15,16]. Most reported ribosomopathies are caused by ribosomal protein gene mutations; however, rDNA alterations could also cause defects in ribosome biogenesis and generate dysfunctional ribosomes.

Only a few studies have examined genetic aberrations in rDNA in lung cancer. Shiao et al. [18] reported the occurrence of single nucleotide polymorphisms (SNPs) in the rRNA gene among human LUAC cell lines in sites located upstream and downstream of the rRNA transcription start site. A recent study demonstrated that coupled 5S rDNA copy number amplification and 45S rDNA loss is associated with the presence of certain somatic genetic alterations, as well as with an increased estimated cancer cell proliferation rate and nucleolar activity across several cancers, including LUAC, gastric adenocarcinoma, and head and neck squamous cell carcinoma [19]. However, the occurrence of germline and somatic DNA alterations in the rDNA promoter region in human LUAC tissues and their clinicopathological relevance have not yet been elucidated. In this study, we examined the frequency of germline and somatic DNA single-nucleotide variants (SNVs) in the rDNA promoter region and their clinicopathological and prognostic relevance in LUAC patients. Our findings suggest the prognostic implications of rDNA promoter genomic alterations in LUAC, and thier potential for future individualized therapies.

## 2. Materials and Methods

### 2.1. Patients and Samples

We retrospectively reviewed 82 LUAC samples that were surgically resected at the Division of Thoracic and Cardiovascular Surgery, Niigata University Medical & Dental Hospital between April 2007 and December 2010. Eligible individuals were required to receive complete resection (surgical margin negative), have pathologically confirmed primary lung cancer, and have a sufficient amount of cancer tissue for genetic analyses. The following clinical data were available: age at surgery, sex, smoking history, Union for International Cancer Control (UICC) tumor-node-metastasis (TNM) classification stage [20], *EGFR* mutation status, and prognosis. *EGFR* mutation testing was performed as previously described [21]. The patients with a missing or too-short follow-up period (less than 30 days) were excluded from the study. Written informed consent for the use of resected tissue for genetic analysis was obtained from each patient. The study was approved by the Ethics Committee on Genetics of Niigata University School of Medicine (G2015-0679, H13-0012) and was conducted in accordance with the Declaration of Helsinki.

The surgically resected specimens were fixed routinely in 10% formalin (4% formaldehyde) solution, and the lung tissues obtained from the maximal cut surface area of the cancer tissues were processed into paraffin blocks for histopathological examination. Tissue sections were prepared (3 μm thickness) and subjected to hematoxylin and eosin (H&E) and Elastica van Gieson staining to diagnose histological subtype and pathological T stage. Histological classification was performed according to the World Health Organization (WHO) classification [22,23]. The pathological stage was determined according to the TNM classification of malignant tumors (UICC) 8th edition [20]. All slides were reviewed by two pathologists (R.O. and H.U.) blinded to the clinical data and sequence results. If the two pathologists had differing opinions regarding the diagnosis, they observed the slides together, and consensus was reached through discussion.

### 2.2. Cell Culture

The human non-small cell lung carcinoma cell lines PC9, NCI-H23, NCI-H441, NCI-H1650, NCI-H2935 and HCC4006, and colon adenocarcinoma cell lines Caco-2, SW480 and COLO205, were purchased from American Type Culture Collection (ATCC; Manassas, VA, USA). The human LUAC cell line A549, colon adenocarcinoma cell lines HCT116 and DLD-1, gastric adenocarcinoma cell lines Kato III and MKN45, Burkitt lymphoma cell line Raji, urinary bladder carcinoma cell line T24, hepatoblastoma cell line HepG2, and cervical carcinoma cell line HeLa were obtained from the Japanese Collection of Research Bioresources (JCRB) Cell Bank (Osaka, Japan). Jurkat cells, a human T cell leukemia cell line, were obtained from the RIKEN Cell Bank (Tsukuba, Japan). The MKN74 cell line was derived from patients with intestinal type gastric carcinoma [24]. All cell lines were cultured in a suitable medium according to the information on culture conditions provided by the American Type Culture Collection (ATCC), the Japanese Collection of Research Bioresources Cell Bank (JCRB), and the RIKEN Cell Bank. Cells were maintained at 37 ℃ in humidified air with 5% CO_2_.

The Pol I-specific inhibitor actinomycin D (ActD) (#A9415, Sigma-Aldrich, St. Louis, Missouri, USA) was initially dissolved in dimethyl sulfoxide (DMSO) (Sigma-Aldrich) and then diluted in the suitable media. Cells were treated with a low concentration of actinomycin D (50 ng/mL) for 2 h at 37 ℃ in humidified air with 5% CO_2_. As a negative control, supplementation with an equivalent amount of DMSO was used.

### 2.3. Microdissection and DNA Extraction

Formalin-fixed paraffin -embedded (FFPE) samples of lung cancers were enriched for neoplastic cellularity to a minimum of 70% of three consecutive 10 μm sections by manual microdissection or laser capture microdissection with a Leica LMD 7000 (Leica Microsystems GmbH, Wetzlar, Germany). For laser capture microdissection, the FFPE sections were placed onto polyethylene naphthalate membrane slides (Leica Microsystems GmbH, Wetzlar, Germany). Microdissected samples were deparaffinized using xylene and 100% ethanol or Deparaffinization Solution (Qiagen, Hilden, Germany) according to the manufacturer’s protocol. For paired normal tissue DNA preparation, lung tissue or resected lymph nodes without cancer cells were used.

DNA from FFPE samples was isolated using the QIAamp DNA FFPE Tissue Kit (Qiagen) on a QIACube (Qiagen) according to the manufacturer’s instructions with the following modifications. FFPE samples were incubated in buffer ATL with proteinase K for 3–6 h at 56 ℃ with occasional vortexing pulses. Genomic DNA was eluted in a 30 μL volume. DNA from cultured cells was extracted using the QIAamp DNA Mini Kit (Qiagen) according to the manufacturer’s instructions. The isolated DNA in the 1.5 mL tube was stored at −20 ℃ until assayed. Before PCR, the isolated DNA was diluted with sterile nuclease-free distilled water (1:10 *v*/*v*).

### 2.4. Sanger Sequencing

The specific primer pairs for 45S rDNA (GenBank Accession No. U13369 and AL592188) and sizes of the expected amplicons are presented in Appendix A. Primers were synthesized by Sigma Aldrich (St. Louis, MO, USA). PCR conditions were as follows: the reaction mixtures contained 5 μL of 5X Colorless GoTaq^®^ Flexi Buffer, 1 μL of MgCl_2_ (25 mM), 0.5 μL of dNTPs (10 mM), 0.5 μL of each primer (10 μM), 0.125 μL of GoTaq^®^ Flexi DNA polymerase (Promega, Madison, WI, USA), 2 μL of diluted (1:10 *v*/*v*) template DNA, and 15.375 μL of sterile nuclease-free distilled water in a total volume of 25 μL. All PCR reactions were carried out as follows: 1 cycle at 95 ℃ for 10 min, followed by 40 cycles each consisting of 95 ℃ for 1 min, 55 ℃ for 1 min, 72 ℃ for 1 min, and a final cycle at 72 ℃ for 10 min. Five microliters of the amplified products was resolved on 2% TBE agarose gels and visualized by ethidium bromide staining. PCR products were purified using ExoSAP-IT (Affymetrix, Cleveland, OH, USA) and subsequently directly sequenced with the BigDye Terminator v1.1 Kit (Applied Biosystems, Foster City, CA, USA). The sequencing products were purified using the NucleoSEQ Kit (MACHEREY-NAGEL, Düren, Germany) and analyzed on a 3500 Genetic Analyzer (Applied Biosystems). The sequence data were analyzed using the GENETYX (Software Development Co. Ltd., Tokyo, Japan) and Chromas software (version 2.6.1; Technelysium Pty, Ltd., South Brisbane, QLD, Australia).

### 2.5. Deep Sequencing Using Illumina MiSeq

Forward (5′-TCGTCGGCAGCGTCAGATGTGTATAAGAGACAG) and reverse (5′-GTCTCGTGGGCTCGGAGATGTGTATAAGAGACAG) Illumina adapter overhangs were added to the 5′ ends of the primers listed in Appendix A to enable Illumina Nextera XT DNA indexing of the PCR products. Primers were synthesized by FASMAC Co. Ltd. (Kanagawa, Japan). Primary PCR was carried out using the GoTaq^®^ Flexi DNA polymerase (Promega) under the same conditions as for Sanger sequencing. Secondary PCR was performed with the Nextera XT Index Kit (Illumina, San Diego, CA, USA) and KAPA HiFi polymerase (Kapa Biosystems, Wilmington, MA, USA). The amplicons were purified after both PCR steps using Agencourt AMPure XP beads (Beckman Coulter, Brea, CA, USA). To confirm the amplicon size and determine the concentration, DNA separation was performed using the DNA 1000 Lab-on-a-Chip Assay Kit with an Agilent Bioanalyzer 2100 (Agilent Technologies, Santa Clara, CA, USA) following the manufacturer’s protocol. The resulting library was sequenced on the Illumina MiSeq with the PhiX Control Kit and MiSeq Reagent Kit Nano v2 (300 cycles) (Illumina). Alignment to the GRCh38 genome was performed using MiSeq Reporter Software v2.5. Integrative Genomics Viewer version 2.3 (Broad Institute, Cambridge, MA, USA) [25] was used to visualize the read alignment. A sequencing coverage of 100× (bidirectional) and a minimum VAF of 5% in the wild-type background were used as cutoffs.

### 2.6. Quantitative Real-Time Reverse Transcription Polymerase Chain Reaction (qRT-PCR)

Total RNA was extracted from PC9, A549, NCI-H441, NCI-H1650, MKN45 and MKN74 using the PureLink™ RNA Mini Kit with PureLink™ DNase Set (Thermo Fisher Scientific, Waltham, MA, USA). Reverse transcription was performed using the SuperScript™ IV VILO™ Master Mix with ezDNase (Thermo Fisher Scientific). Quantitative real-time PCR was performed using SsoAdvanced™ Universal SYBR Green Supermix (Bio-Rad, Hercules, CA, USA) on a LightCycler^®^ 96 Real-Time PCR System (Roche, Basel, Switzerland). Sequences of the primer sets are shown in Appendix A. The amplification conditions were as follow: 94 °C for 3 min, followed by 45 amplification cycles at 94 °C for 15 s, 60 °C for 20 s and 72 °C for 30 s. Relative fold changes were determined by the comparative threshold (ΔΔCT) method using cyclophilin as endogenous normalization control.

### 2.7. Statistical Analysis

All statistical analyses were performed using R version 3.4.1 (R Foundation for Statistical Computing, Vienna, Austria) and EZR version 1.37 (Saitama Medical Center, Jichi Medical University, Saitama, Japan) [26]. Statistical evaluation of the associations between two categorical variables was based on Fisher’s exact test. In the qRT-PCR analysis, significant differences were calculated from the average value of 3 wells. Data are presented as mean ± SD and the statistical analysis involved *p*-values calculated by paired two-tailed Student’s t tests. Kaplan–Meier survival curves were constructed for recurrence-free survival (RFS) and overall survival (OS). Differences between the groups were estimated using the log-rank test. From univariate analysis, we selected variables with *p*-value < 0.05 (statistical criterion) by Cox regression and assessed multicollinearity between variables. A multivariate analysis was undertaken through Cox regression with Firth’s penalized likelihood [27,28]. All statistical tests were two-sided; a *p*-value < 0.05 was regarded as statistically significant.

## 3. Results

### 3.1. Assessment and Validation of the Assay Design for Identification of rDNA Target Region

The human rRNA promoter contains two essential elements: the upstream control element (UCE; −156 to −107) and the core promoter (CP; −45 to +18). The UCE serves as the binding target for cellular trans-activating proteins, such as upstream binding factor (UBF), and is involved in transcription initiation of Pol I, whereas the CP overlaps the transcription start site (+1). The entire promoter region of rDNA is contained in an intergenic spacer region (IGS) between rDNA units, which are transcribed as long precursors known as 45S precursor rRNA (45S pre-rRNA) that are subsequently rapidly spliced into the 18S, 28S, and 5.8S rRNA transcripts [11,12,13,15,16,17,29,30].

We designed primers to cover three target regions in rDNA (Figure 1 and Appendix A): (i) the rDNA promoter from position −248 to +100, which partially overlaps with that previously analyzed in another study [31]; (ii) the 18S rRNA coding region from position +5244 to +5324; and (iii) the 28S rRNA coding region from position + 12,320 to +12,400 (nucleotide positions are in reference to GenBank Accession No. U13369).

In total, 20 human cancer cell lines were screened by both a PCR-based targeted next-generation sequencing (NGS) method and Sanger sequencing. The SNVs identified by NGS with a minimum variant allele frequency (VAF) of 5% are summarized in Table 1. Recurrent SNVs at positions −206, −96, −72, and +49 were observed in more than one cell line, respectively. Other SNVs were observed in only one of the cell lines (Figure 2 and Appendix A, and Table 1). None of the cells had SNVs in rDNA coding regions, including 18S rRNA or 28S rRNA.

Next, we compared the Sanger sequencing and NGS results. Generally, recent descriptions of NGS analysis pipelines for laboratory testing for detecting alleles of clinical samples have recommended using a detection threshold of 5% to minimize technical artifacts and give reliability [32]. When a minimum VAF of 5% was used as the cutoff, all variants found in all cultured cells demonstrated 100% concordance between Sanger sequencing and NGS (Figure 2 and Appendix A). Based on the data obtained from the cell lines, we decided to focus on the rDNA promoter region for subsequent analyses of cancer tissue samples using the more cost-effective Sanger sequencing.

### 3.2. Patient Characteristics

The demographic and clinicopathological characteristics of 82 primary LUAC patients are listed in Table 2. Histologically, there were 8 (9.8%) minimally invasive adenocarcinoma (MIA) cases, 3 (3.7%) lepidic predominant adenocarcinoma, 8(9.8%) acinar predominant adenocarcinoma, 33 (41.5%) papillary predominant adenocarcinoma, 2 (2.4%) micropapillary predominant adenocarcinoma, 15 (18.3%) solid predominant adenocarcinoma, 8 (9.8%) invasive mucinous adenocarcinoma, 2 (2.4%) fetal adenocarcinoma, and 2 (2.4%) pleomorphic carcinoma (1 was acinar predominant and 1 was solid predominant). In total, 3 cases (2 of 34 papillary and 1 of 8 acinar predominant) fulfilled the criteria of adenosquamous carcinoma; adenocarcinoma with focal squamous cell carcinoma component (more than 10% and less than 40%). There was no adenocarcinoma in situ case (Table 2). The median follow-up period for surviving patients was 113.9 months (range, 10.7–145.9).

### 3.3. Germline and Somatic SNVs in the rDNA Promoter Region Among 82 LUAC Patients

Seven loci of gene alterations, including positions −206, −204, −96, −72, +49, +52 and +73, in the rDNA promoter region were identified in 35 cases (42.7%); 29 cases (35.4%) carried germline SNVs, 8 cases (9.8%) carried somatic SNVs, and both germline and somatic SNVs were detected in 2 cases (2.4%). Recurrent gene alterations were observed at positions −206, −96, −72, +49, and +52. Among 29 LUACs with germline SNVs, 9 (31.0%) were located at position −206, 1 (3.4%) at position −204, 8 (27.6%) at position −96, 2 (6.9%) at position +49, and 10 (34.5%) at position +52. Among eight LUACs with somatic SNVs, three (37.5%) were detected at position −96, three (37.5%) at position −72, and two (25.0%) at position +51 (Figure 3, Figure 4 and Figure 5 and S2, Table 3 and Appendix A).

The presence of germline and/or somatic SNVs at any site (from position −248 to +100) from position −248 to +1, and germline SNV at any site (from −248 to +100), was associated with older age (*p* = 0.002, *p* = 0.016, and *p* = 0.007, respectively). The frequency of germline and/or somatic SNVs at position −96 was higher in males (*p* = 0.031). No association was found between other germline and/or somatic SNVs and any other clinicopathological parameters, including sex, age, smoking history, stage, nodal and/or distant metastasis, *EGFR* mutations, and MIA versus invasive histological subtype.

### 3.4. Survival Analysis of LUAC Patients

The 5-year and 10-year RFS rates in LUAC patients were 63.9% and 59.1%, respectively; the 5-year and 10-year OS rates were 79.1% and 59.1%, respectively.

The presence of germline SNVs at position +1 to +100 in the rDNA promoter region, lesions of higher stages (III and IV), nodal and/or distant metastasis, and invasive histological subtypes was significantly associated with worse prognosis in terms of both RFS (Figure 6A and Appendix A; *p* = 0.030, *p* < 0.001, *p* < 0.001, and *p* = 0.022, respectively) and OS (Figure 7A and Appendix A; *p* = 0.004, *p* < 0.001, *p* < 0.001, and *p* = 0.032, respectively) after surgery. Germline SNVs at position +49 and +52 were significantly associated with shorter OS (*p* = 0.001 and *p* = 0.035, respectively), whereas no significant association with RFS was detected (Appendix A). None of the somatic SNVs had a significant correlation with survival. Although germline and/or somatic mutations at sites extending from position +1 to +100 were associated with worse OS (*p* = 0.025), the association did not reach statistical significance for RFS. No association was found between clinical outcome and other clinicopathological factors, including sex, age, *EGFR* mutation, smoking, or presence of other types of mutations (Appendix A).

The results of univariate and multivariate Cox regression analyses are shown in Table 4. The variable nodal and/or distant metastasis was excluded from the final model to avoid collinearity with stage. In multivariate Cox regression analysis, the presence of germline rDNA promoter SNVs between position +1 and +100 was an independent prognostic factor for unfavorable RFS (hazard ratio (HR), 3.669; 95% confidence interval (CI), 1.545–8.373, *p* = 0.004) and OS (HR, 4.870; 95% CI, 2.004–11.56, *p* < 0.001) after adjustment for age, pathologic stage, and invasive/MIA histological subtype. From the results of the multivariate analysis, we further evaluated the prognostic impact of the presence of germline rDNA promoter SNVs between position +1 and +100 in Stage I and Stage I/II patients after excluding MIA cases (*n* = 42 and *n* = 60, respectively). The presence of germline SNVs between position +1 and +100 was significantly associated with a worse prognosis for RFS (*p* = 0.004 in Stage I; *p* = 0.001 in Stage I/II) and OS (*p* < 0.001 in Stage I and Stage I/II) in both subgroups by log-rank test (Figure 6B,C and Figure 7B,C).

### 3.5. rDNA Promoter SNVs in Publicly Available Databases

To validate the prognostic value of rDNA promoter SNVs, we searched the publicly available germline and somatic variant databases derived from whole-exome sequences (WXS) and whole-genome sequences (WGS), including the International Cancer Genome Consortium (ICGC) [33], the National Bioscience Database Center Human Database [34], the European Genome-phenome Archive [35], the Cancer Genome Atlas (TCGA) [36], and the Genome Aggregation Database (gnomAD) [37]. In the ICGC data portal [33], there were two sets of open-access data comprised of somatic variants. GRCh37 (hg19) was the reference genome used for the alignment. In hg19 (National Center for Biotechnology Information; NCBI), 45S rDNA was unplaced, unlocated, and unlocalized, and it was newly mapped in GRCh38 (RNA45SN1, N2, N3, N4, N5 in NCBI Gene). Therefore, the 45S rDNA sequence has not been analyzed in the ICGC. In the TCGA data portal [36], the TCGA lung adenocarcinoma (TCGA-LUAD) open-access data comprised only somatic variants by using GRCh38 as the reference genome. TCGA-LUAD is a WXS dataset which was sequenced using the Agilent SureSelect Human All Exon 50Mb kit (Agilent Technologies, Palo Alto, CA, USA) [38,39]. We obtained the BED file for the kit containing the coordinates of capture target genomic positions from Agilent. The original BED file was designed based on hg19, and was converted from the GRCh37/hg19 genome assembly to GRCh38/hg38 with UCSC Genome Browser’s LiftOver [40], using the default webtool parameters according to the manufacturer’s instruction. We confirmed that TCGA-LUAD WXS did not analyze the rDNA gene promoter regions (positions −248 to +100 of RNA45SN1 at chromosome (chr) 21: 8432974-8433321, RNA45SN2 at chr21:8205740-8206087, RNA45SN3 at chr21:8388787-8389134) by design. The gnomAD is a large population genomics database which compiles germline WGS and WXS studies around the world [37]. One of its datasets, the gnomAD v2.1, contains data from 125,748 exomes and 15,708 whole genomes, including the Exome Aggregation Consortium (ExAC) dataset from blood (“germline”) samples from TCGA. However, the gnomAD v2.1 was based on hg19. Another dataset, gnomAD v3, was based on GRCh38 and includes 71,702 WGS data. The ribosomal RNA gene promoter region showed “No coverage”, i.e., the sequence of this region was not read in all WGS studies presented in gnomAD v3 (Appendix A). Thus, we were not able to find suitable information that can be used to validate our findings obtained from our patient cohort.

### 3.6. Evaluation of rRNA Expression

To investigate whether the rDNA promoter SNVs between position +1 and +100 influence rRNA synthesis, we performed qRT-PCR to compare rRNA transcription levels between the LUAC cell lines carrying the SNVs and those without the SNVs. PC9 cells with the SNV at +49, A549 cells with the SNV outside +1 to +100, and the H1650 cells without the SNVs were treated with vehicle control (DMSO) or low concentrations of ActD (50 ng/mL), which has been shown to selectively inhibit the Pol I-driven rRNA transcription [41,42]. As shown in Figure 8, the PC9 and A549 cells showed significantly higher transcription levels of endogenous 47/45S pre-rRNA than H1650 (*p* < 0.05), whereas there was no significant difference in 47/45S pre-rRNA transcripts between PC9 and A549 cells. No significant differences were observed in endogenous 28S mature rRNA transcription levels between the cells with SNV and those without SNVs. When the cells were treated with low-dose ActD, a dramatic decrease in the level of 47/45S pre-rRNA was observed in all the three cell lines (*p* < 0.05). As for 28S mature rRNA expression, ActD did not affect the 28S mature rRNA transcription level in PC9. In A549 and H1650 cells, ActD caused a mild to moderate reduction in 28S mature rRNA, but the difference was not significant (*p* = 0.099 and *p* = 0.053, respectively). ActD-treated PC9 with the SNV at +49 retained significantly higher 47/45S pre-rRNA and 28S mature rRNA expressions compared to ActD-treated H1650 without SNV (*p* < 0.05).

## 4. Discussion

In this study, we investigated the clinical and prognostic features of germline and somatic single nucleotide genetic alterations in the rDNA promoter region among Japanese LUAC patients. Our study identified frequent and recurrent genomic alterations in the rDNA promoter region in normal and LUAC tissues. Importantly, the presence of germline SNVs between positions +1 and +100 in the rDNA promoter region demonstrated a significant association with worse prognosis in terms of both RFS and OS.

Recent technical advances in large-scale sequencing and genomics methods have enabled the comprehensive analysis of somatic SNVs in human long ncRNAs (lncRNAs), impacting lncRNA expression [43,44], and studies have been conducted to unveil the role of lncRNAs in the regulation of rRNA synthesis in human cancer [45,46]. However, few studies have focused on the existence of SNVs and/or SNPs in human rDNA and their functional role in the regulation of rRNA synthesis. Shiao et al. identified hotspot SNPs at positions −234, −233, −181, −104, −96, −72, +52, +139, +144, +207, +225 and +290 (relative to the transcription start site at position +1) using sequencing products amplified from the region −388 to +306 of the rRNA in six human LUAC cell lines and a nontransformed, immortalized line from human peripheral lung epithelium. Importantly, no SNP was identified in the UCE or CP [18]. In our work, several SNVs were detected in the rDNA promoter region in the A549, H23, PC9, Caco-2, SW480, COLO205, DLD-1, Raji, Jurkat, and HeLa cell lines. Consistent with the previous study [18], the SNV at position −96 was found in the A549 LUAC cell line, and no SNVs were detected in PCR-amplified products from H441 or H23 cell lines. As expected, all 20 cell lines lacked SNVs in UCE or CP regions.

Given the unique array of rDNA genes with several hundreds of transcription units organized in tandem repeats and clustered on five chromosomal loci [11], it might be challenging to reliably detect variants with low VAF using the Sanger method if the primers bind to the promoters of all rDNA genes on different chromosomes. For the assessment of the whole population of molecules and validation of the results obtained using the Sanger method, pyrosequencing was utilized by Shiao et al. [18], while we used NGS, which is the more recent revolutionary and reliable method. Although recent advances in NGS employing unique molecular identifiers have improved the sequencing accuracy, the current inherent error rates of NGS library preparation approaches, sequencing chemistry and platforms complicate the reliable detection of low-frequency variants without compromising specificity. Generally, the lower limit of VAF that has clear clinical utility is generally in the range of 5% [32]. Considering a minimum VAF cutoff of 5% in NGS, the sequence results detected in all cultured cells showed 100% agreement between Sanger and NGS data. These results support the reproducibility of our assay, despite the complicated repetitive genomic structure of the rDNA genes [11]. While we focused on the significant common variants with ≥5% VAF in this study, recent genetic studies have shown the importance of the low-frequency variants with less than 5% VAF in the pathogenesis of diseases, such as somatic mutations in cancer [47], genetic disorders [48], as well as the hereditary risks of diseases and their progression [49,50], including lung cancer [51]. Sequencing methods and bioinformatics pipelines that enable the distinction of true variants from artifacts are improving [50,51,52,53,54]. In comparison to oncogenes and tumor suppressor genes (2 copies), the interpretation of VAFs thresholds for rDNA (hundreds of copies) is much more challenging. Future advances in sequencing technology may allow the accurate identification of low-frequency pathogenic variants in the rDNA genes, and the understanding of their extensive role in lung cancer.

Of note, Shiao et al. also found that differences in the frequency of C residue at position +139, which is located downstream of the rRNA transcription start site, correlated negatively with rRNA abundance [18], albeit relying on a limited data set from a small number of cell lines. Another study by Zhang et al. [31] investigated the frequency of SNPs at CpG sites in the region surrounding the transcription start site for rRNA (from position −248 to +100) in a human keratinocyte cell line, HaCaT. They also demonstrated that the binding site of Pol I regulatory proteins such as UBF and basonuclin is located downstream and upstream of UCE and CP, respectively, which were initially identified as Pol I regulatory sites by their association with subunits of Pol I. The CpG island methylation level at the promoters of tumor suppressor genes is widely known to contribute to tumorigenesis and tumor progression in many types of cancer by changing the promoter activity [55]. The methylation status of the rDNA promoter region and its relationship with 45S pre-rRNA expression have been evaluated in several types of tumors, including breast, colon, oral, prostate, uterine, and cervical cancers, and myelodysplastic syndrome, but the results vary among tumor types [56,57,58,59,60,61]. In the only study available (to our knowledge) on the effect of CpG site-specific methylation status on promoter activity, Ghoshal et al. [62] reported that most CpGs between positions −136 and −58 were significantly hypomethylated, and the methylation of the rDNA promoter region resulted in a decline in the expression of rRNA genes in human hepatocellular carcinoma. The methylation of CpG sites at −347, −102 and −9 inhibited rRNA promoter transcription activity, probably by recruiting methyl-CpG binding proteins, whereas the CpG site at +152 did not play a role in the promoter activity. These observations suggest that a much broader region than UCE and CP may serve as the rDNA promoter, but whether the methylation status at specific CpG sites in the rDNA coding region has an effect on rDNA promoter activity is still unclear.

Previous studies have identified hypermethylation of the rDNA promoter region as a prognostic factor in breast [63], endometrial [64], and ovarian cancers [65]. The increased expression of 45S pre-rRNA is linked to the poor prognosis of patients with rhabdomyosarcoma [66] and colon cancer [67]. In this study, recurrent rDNA promoter SNVs at positions −206, −96, and −72 alone had no significant prognostic impact regardless of whether they were germline or somatic alterations, whereas germline SNVs at positions +49 and +52 alone, as well as the germline alterations found at position +1 to +100, were significantly associated with worse prognosis. In particular, the single nucleotide alteration at position +49 observed in CpG sites may affect the methylation of the rDNA promoter. The correlation between 45S pre-rRNA expression levels and rDNA promoter SNVs in LUAC and corresponding non-tumor tissues was not assessed because the RNA quality was not consistent among FFPE tissue samples due to degradation. Such an analysis would be valuable because it could provide functional insights into the role of rDNA promoter SNVs.

Instead, to determine the possible effect of the rDNA promoter SNVs on ribosomal biogenesis and under ribosomal stress (also called nucleolar stress) [68], we used low-dose ActD to inhibit ribosome biogenesis in LUAC cell lines. Interestingly, the endogenous 47/45S pre-rRNA transcription level was significantly higher in PC9 cells carrying the rDNA promoter SNV, and A549 with the SNV outside the range of +1 to +100, than H1650 without SNV, although low-dose ActD significantly reduced the 47/45S pre-rRNA synthesis and no significant decrease in the 28S mature rRNA level was observed in all three cell lines. PC9 cells carrying the rDNA promoter SNV at +49 retained a transcription level of 47/45S pre-rRNA and 28S mature rRNA that was significantly higher than H1650 without the SNV. These results suggest that the rDNA promoter SNVs may be involved in ribosomal biogenesis, ribosomal/nucleolar stress tolerance, drug resistance and cell proliferation. A549 cells with the SNV outside the range +1 to +100 did not retain 45S pre-rRNA and 28S mature rRNA expression as much as PC9, but showed a mild decrease in 28S mature rRNA compared to H1650, suggesting that the ribosome biogenesis rate and the effect of ActD may differ depending on the location of SNVs. Overall, 28S rRNA production was less affected by ActD than de novo rRNA production represented by 47/45S pre-rRNA. This was expected because 28S rRNA is much more stable to low-dose ActD than 47/45S pre-rRNA, but can be reduced in several cell lines depending on the ActD concentration and treatment time [69,70,71,72]. Further validation experiments using multiple cell lines are warranted to confirm the suppressive effect of the SNV on ribosomal biogenesis and ribosomal/nucleolar stress. Ribosome biogenesis is a complex metabolic process that involves a large number of molecules, including ribosomal proteins [11,14,16,17], ncRNAs [11,73], UBF [14,15], and other suggested Pol I regulatory factors, such as Runx2 [74] and RasL11a [75]. Additionally, ribosomal stress response is regulated by numerous molecules in both p53-dependent and p53-independent manners [76,77]. Whether the SNV in the PC9 cell line is somatic or germline remains unknown. It will be important and informative to investigate the p53 and UBF phosphorylation status, the methylation status of the rDNA promoter, and the affinity of UBF and other regulatory factors to the rDNA promoter region in cells carrying SNVs and those without SNVs via chromatin immunoprecipitation (ChIP) and NGS-based ChIP-sequencing. In order to understand why germline rDNA promoter SNVs had more predictive power than somatic SNVs, and to elucidate the particular effects of individual alterations on methylation status and ribosomal biogenesis, will require further investigation.

The prognostic power of somatic SNVs from positions +1 to +100 was equivocal, probably because of their rarity in our data set (*n* = 2, at position +52) (Appendix A and Table 3) and due to the small sample size. Interestingly, germline and somatic rDNA promoter SNVs were nearly mutually exclusive, except in two LUAC cases (Table 2, Table 3, and Appendix A). The limited sample size also caused an unbalanced distribution of some clinical characteristics, such as sex, number of stage I tumors at surgery, and histological subtype of LUAC. In addition, the cohort solely consisted of Japanese patients. In an attempt to confirm the prognostic implications of the rDNA promoter germline SNVs of the patient cohort used in this study, we searched public databases containing the WGS or WXS data of LUAC to validate our results in another independent cohort. Unfortunately, the rDNA promoter SNVs were not available in the open-access data of the WGS or WXS databases, including TCGA [36] and gnomAD [37]. Future large-scale studies are needed to validate the robustness of the SNV signatures in rDNA and to elucidate their prognostic implications.

Importantly, our survival analysis revealed a clear association of the presence of germline SNVs between +1 to +100 in the rDNA promoter region with an adverse outcome, independent of stage and histology, particularly in the patients with pathological early-stage invasive LUAC. Clinically, adjuvant chemotherapy is considered the global standard of care for patients with completely resected NSCLC, including adjuvant tegafur/uracil chemotherapy for stage I NSCLC patients in Japan [78,79], and global standard cisplatin-based adjuvant chemotherapy for the patients at stages II to III of the disease [79,80,81]. Nevertheless, the considerable acute and late toxicities could be associated with adjuvant chemotherapy [79,80,81]. The better identification of the patient subgroups that are at greater risk of disease progression and are more likely to benefit from adjuvant chemotherapy is needed. No biomarker currently exists to predict the effects of adjuvant chemotherapy on OS [81]. If the prognosis could be improved by adding chemotherapy to the poor prognosis group with rDNA promoter germline SNVs, germline rDNA promoter SNV status could be a valuable molecular marker to refine patient selection for adjuvant chemotherapy.

Ribosome biogenesis takes place in the nucleoli, which have long been linked to cancer; an increase in the number and size of nucleoli has been used as a marker of malignancy for over a century [14,82,83,84]. Based on the finding that an increase in rRNA transcription affects cancer progression by not only promoting protein synthesis, and thus proliferative capacity, but also controlling cellular checkpoints and chromatin structure, targeted therapeutic approaches with first-in-class selective inhibitors of rDNA transcription, CX-5461 and PMR-116, have been explored [85,86]. The in vitro screening of the cytotoxic activity of CX-5461 revealed a broad spectrum of activity in cell lines derived from various cancers, including the A549 LUAC cell line [87]. In 2018, a phase I clinical trial of CX-5461 for hematological malignancies was completed, and phase I/II trials for solid tumors, including breast cancer and non-melanomatous skin cancer, are underway (Clinical trial identification, NCT02719977) [85,86]. Further analysis is needed to investigate the effects of rDNA promoter SNVs on ribosome biogenesis and prognosis.

## 5. Conclusions

We have identified frequent germline and somatic rDNA promoter SNVs in human LUAC tissues. The presence of the germline rDNA promoter SNVs located in the region between position +1 and +100 (with respect to the transcription start site at +1) was significantly associated with an adverse outcome in our LUAC patient cohort. Additional studies are required to validate our findings with larger cohorts and to investigate the functional role of SNVs at specific sites of the rDNA promoter region in cancer progression.

## Figures and Tables

**Figure 1 cells-09-02409-f001:**
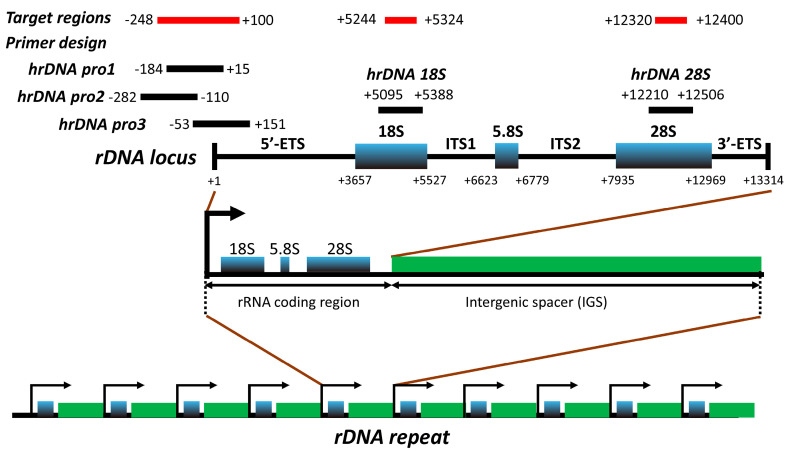
Location of the target regions selected for sequence analysis within the rDNA locus. Base positions are relative to transcription start site (+1) of the rRNA gene (rDNA). Each amplicon corresponds to the amplified strand indicated in Appendix A. 5’-ETS, 5’ external transcribed spacer; ITS1, internal transcribed spacer 1; ITS2, internal transcribed spacer 2; 3’-ETS, 3’ external transcribed spacer.

**Figure 2 cells-09-02409-f002:**
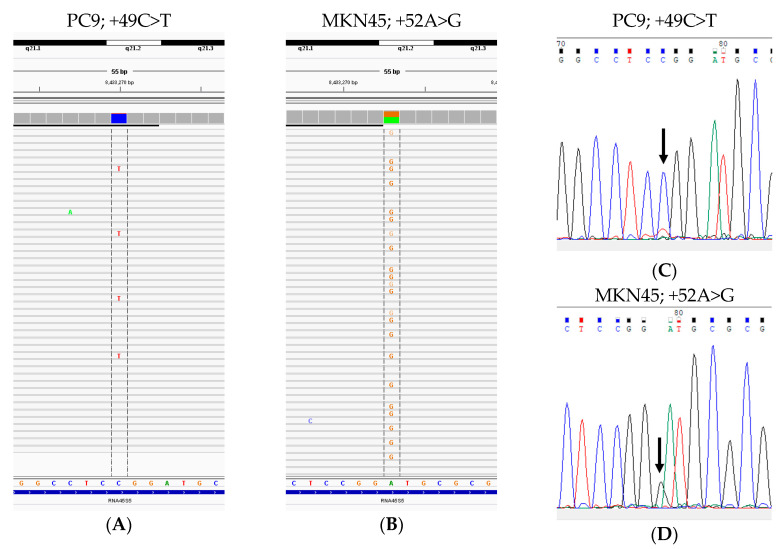
Representative single-nucleotide variations (SNVs) observed in cell lines. (**A**) PC9; +49C>T (variant allele frequency (VAF]): 8%) and (**B**) MKN45; +52A>G (VAF: 47%) detected by next-generation sequencing. (**C**,**D**) Sanger electropherograms depicting SNVs PC9, +49C>T (**C**) and MKN45, +52A>G (**D**).

**Figure 3 cells-09-02409-f003:**
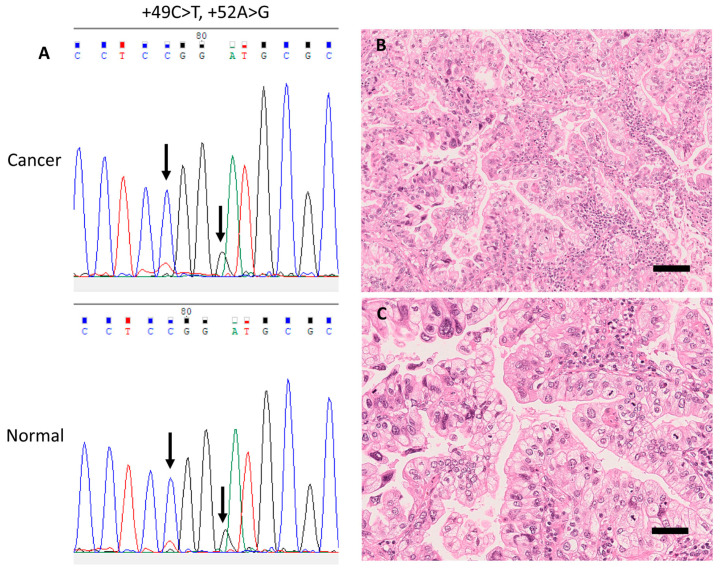
Representative case with a germline single-nucleotide variant (SNV) in the rDNA promoter (case 1). (**A**) Electropherograms of germline SNVs +49C>T and +52A>G from corresponding cancer (upper panel) and normal (lower panel) tissue samples. (**B**) Representative histopathological images of this case (H&E stain) showing invasive mucinous carcinoma accompanied with papillary structure, and (**C**) nuclear pleomorphism and frequent mitoses. Scale bar, 50 μm.

**Figure 4 cells-09-02409-f004:**
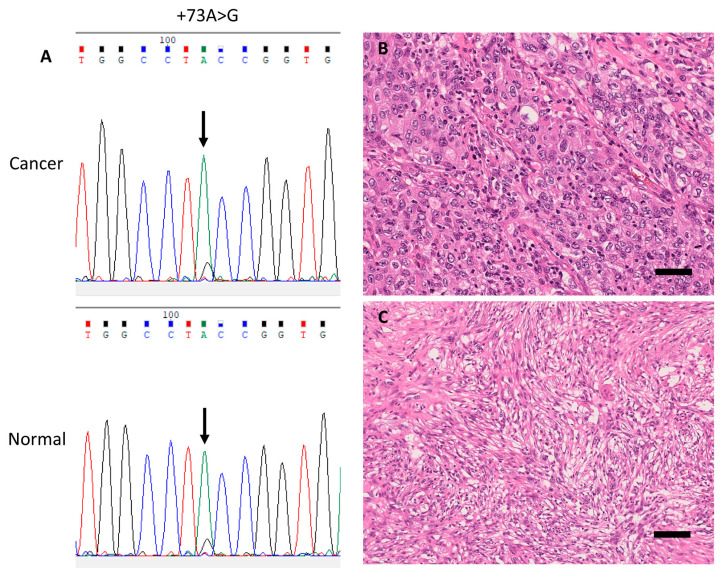
Representative case with a germline single-nucleotide variant (SNV) in the rDNA promoter (case 3). (**A**) Electropherograms of germline SNV +73A>G from corresponding cancer (upper panel) and normal (lower panel) tissue samples. **(B**) Representative histopathological images of this pleomorphic carcinoma case (H&E stain) showing a solid predominant phenotype, and (**C**) sarcomatoid component composed of pleomorphic spindle cells. Scale bar, 100 μm.

**Figure 5 cells-09-02409-f005:**
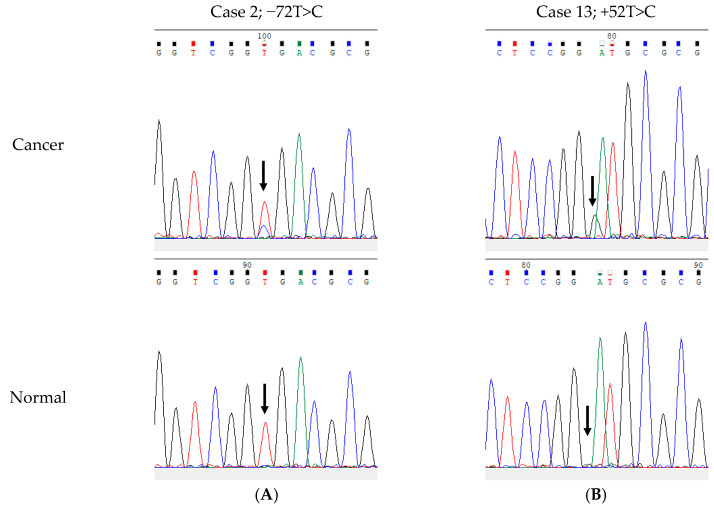
Electropherograms of representative somatic single-nucleotide variants (SNV) in the rDNA promoter region in corresponding cancer tissue (upper panel) and normal tissue (lower panel). (**A**) SNV −72T>C in case 2 and (**B**) +52A>G in case 13. Note that there was no apparent additional peak at +49 in both cancer and normal tissues in this case, suggesting that Case 1 harbored +49 germline SNV.

**Figure 6 cells-09-02409-f006:**
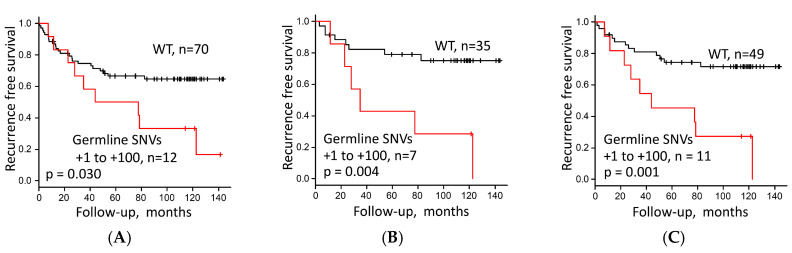
Recurrence-free survival stratified by (**A**) presence or absence of germline single-nucleotide variants (SNVs) at any site at positions +1 to +100 in the rDNA promoter region (presence versus wild type (WT)) in 82 lung adenocarcinoma (LUAC) patients, and (**B**) germline SNV at positions +1 to +100 in the rDNA promoter region (presence versus WT) in Stage I LUAC excluding MIA (*n* = 42), and (**C**) in Stage I/II LUAC excluding MIA (*n* = 60).

**Figure 7 cells-09-02409-f007:**
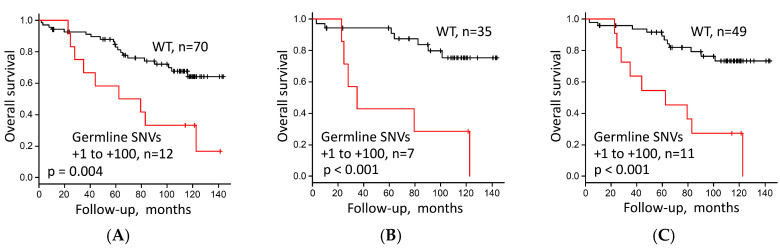
Overall survival stratified by (**A**) germline single-nucleotide variants (SNVs) at any site at positions +1 to +100 in the rDNA promoter region (presence versus wild type (WT)) in 82 lung adenocarcinoma (LUAC) patients, and (**B**) germline SNV at positions +1 to +100 in the rDNA promoter region (presence versus WT) in Stage I LUAC excluding MIA (*n* = 42), and (**C**) in Stage I/II LUAC excluding MIA (*n* = 60).

**Figure 8 cells-09-02409-f008:**
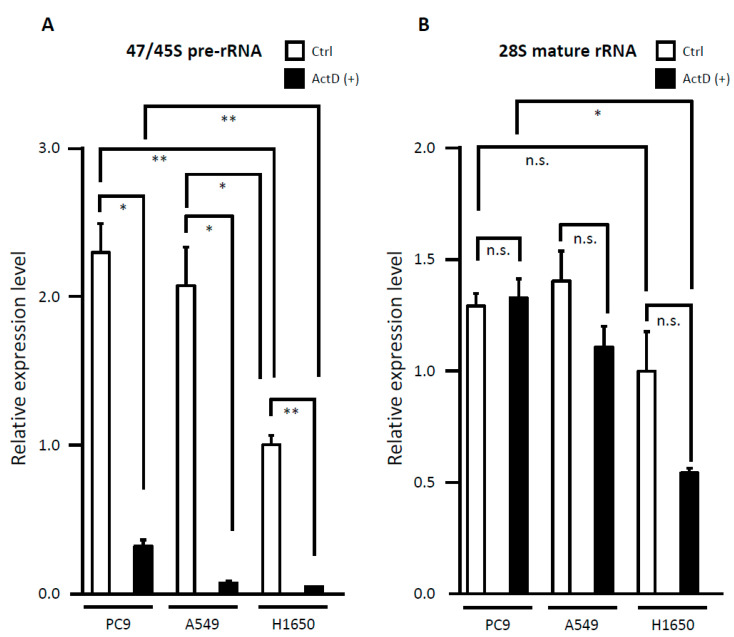
Evaluation of rRNA expression and the effect of low-dose Actinomycin D (50 ng/mL, 2 h) treatment in lung adenocarcinoma cell lines. (**A**) 47/45S precursor ribosomal RNA (pre-rRNA) and (**B**) 28S mature rRNA. Relative transcription level of untreated H1650 cells was assumed as 1. n.s., not significant; * *p* < 0.05; ** *p* < 0.01.

**Table 1 cells-09-02409-t001:** SNVs around rDNA promoter region in 20 human cancer cell lines.

Cell lines	Tissue of Origin	Histological Type	SNVs ^1^
A549	Lung	Adenocarcinoma	−104T>C, −96C>T
H23	Lung	Adenocarcinoma	−210T>C, −206C>T
PC9	Lung	Adenocarcinoma	+49C>T
Caco-2	Colon	Adenocarcinoma	−186G>C
SW480	Colon	Adenocarcinoma	−96C>T
COLO205	Colon	Adenocarcinoma	−96C>T, −79T>A
MKN45	Stomach	Adenocarcinoma	+52A>G
DLD-1	Colon	Adenocarcinoma	−72T>C
Raji	B-lymphocytes	Burkitt’s lymphoma	−96C>T, −85G>C, +49C>T
Jurkat	T-lymphocytes	T cell acute lymphoblastic leukemia	−96C>T
HeLa	Uterine cervix	Adenocarcinoma, HPV18-positive	−206C>T, −96C>T, −72T>C

^1^ SNV, single nucleotide variant.

**Table 2 cells-09-02409-t002:** Patient characteristics.

Characteristics	Group	*n* (%)
No. of patients		82
Gender	Female	26 (31.7)
	Male	56 (68.3)
Age: Median (range), y		66.5 [33,34,35,36,37,38,39,40,41,42,43,44,45,46,47,48,49,50,51,52,53,54,55,56,57,58,59,60,61,62,63,64,65,66,67,68,69,70,71,72,73,74,75,76,77,78,79,80,81,82,83,84,85,86,87]
	≤55	11 (13.4)
	>55	71 (86.6)
Smoking history	Never	31 (37.8)
	Former/current	51 (62.2)
Stage	I	50 (61.0)
	II	18 (22.0)
	III	11 (13.4)
	IV	3 (3.7)
Nodal and/or distant metastasis	Present	19 (23.2)
	Absent	63 (76.8)
*EGFR* mutation	Present	25 (30.5)
	Exon 19 deletion	14 (17.1)
	Exon 21 L858R	11 (13.4)
	Exon 20 insertion	0 (0)
	Exon 18 G719S	0 (0)
	Absent	49 (59.8)
	NA	8 (9.8)
Histology	MIA	8 (9.8)
	Lepidic	3 (3.7)
	Acinar	8 (9.8)
	Papillary	34 (41.5)
	Micropapillary	2 (2.4)
	Solid	15 (18.3)
	IMA	8 (9.8)
	Fetal	2 (2.4)
	Pleomorphic	2 (2.4)
SNV in the rDNA promoter region	Germline	29 (35.4)
	Somatic	8 (9.8)
	Germline and somatic (at different locus)	2 (2.4)
	Total (Germline and/or somatic)	35 (42.7)
	No SNV	47 (57.3)

NA, not assessed; MIA, minimally invasive adenocarcinoma; IMA, invasive mucinous adenocarcinoma; SNV, single-nucleotide variant; rDNA, ribosomal RNA gene, ribosomal DNA; *n*, number of patients.

**Table 3 cells-09-02409-t003:** rDNA promoter SNVs (at positions +1 to +100) found in lung adenocarcinoma patients.

Patient	Histology	Sex	Age	Stage	Germline	Somatic	Months Post-Surgery	Alive/Deceased
1	IMA	F	67	IA2	−206 ^1^, +49, +52		35.3	Deceased
2	Acinar	M	78	IIB	+49	−72 ^1^	44.5	Deceased
3	Pleomorphic + Solid	M	68	IIA	+73		63.3	Deceased
4	Acinar	M	63	IB	+52		124.5	Deceased
5	Papillary	F	74	IIB	+52		84.1	Deceased
6	Acinar	M	71	IA2	+52		80.5	Deceased
7	Acinar (ASQ)	M	67	IB	+52		24.8	Deceased
8	Papillary	M	79	IA	+52		22.9	Deceased
9	Papillary	F	71	IA3	+52		28.1	Deceased
10	MIA	F	67	IA1	+52		143.4	Alive
11	Papillary	M	62	IB	+52		123.1	Alive
12	Solid	M	66	IIB	+52		115.6	Alive
13	Papillary	F	57	IB		+52	145.9	Alive
14	Papillary	M	79	IVB		+52	50.7	Alive

rDNA, ribosomal RNA gene, ribosomal DNA; SNV, single-nucleotide variant; IMA, invasive mucinous adenocarcinoma; ASQ, adenosquamous carcinoma; MIA, minimally invasive adenocarcinoma; M, male; F, female; Stage, UICC TNM stage. ^1^ SNVs outside +1 to +100 were also detected in these cases.

**Table 4 cells-09-02409-t004:** Univariate and multivariate Cox regression analyses of recurrence-free survival and overall survival of lung adenocarcinoma patients.

Variable	Univariate			Multivariate ^2^		
	HR	95% CI	*p*	HR	95% CI	*p*
**Proportional hazard model for recurrence-free survival**						
Age (>55/≤55)			n.s.			
Sex (male/female)			n.s.			
Smoking status (Former/never)			n.s.			
pN/N and/or cM (>1/0) ^1^	3.748	1.813–7.747	<0.001 *	-	-	-
Stage (III–IV/I–II)	4.802	2.208–10.45	<0.001 *	6.106	2.587–14.08	<0.001 *
Histology (Invasive/MIA) ^2^	9.212	1.303–1166.69	0.019 *	7.267	1.006–923.71	0.049 *
rDNA promoter SNV (Present/Absent)	2.294	1.060–4.964	0.035 *	3.764	1.584–8.589	0.004 *
**Proportional hazard model for overall survival**						
Age (>55/≤55)			n.s.			
Sex (male/female)			n.s.			
Smoking status (Former/never)			n.s.			
pN/N and/or cM (>1/0) ^1^	3.607	1.690–7.701	<0.001 *	-	-	-
Stage (III–IV/I–II)	3.455	1.566–7.621	0.002 *	4.745	1.951–11.27	<0.001 *
Histology (Invasive/MIA) ^2^	8.097	1.140–1206.18	0.031 *	7.192	0.985–915.93	n.s.
rDNA promoter SNV (Present/Absent)	2.995	1.360–6.596	0.006 *	5.006	2.060–11.88	<0.001 *

HR, hazard ratio; CI, Confidence interval; Stage, UICC TNM stage; MIA, minimally invasive adenocarcinoma; SNV, single-nucleotide variant, rDNA promoter SNV, germline SNV from the position +1 to +100; n.s., not significant; * *p* < 0.05. ^1^ This variable was excluded from the final multivariate Cox regression model to avoid multicollinearity in relation to the Stage (III–IV/I–II). ^2^ Firth’s correction was used because of quasi-complete separation; there was no event in one of the subgroups.

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
