# Peer review of "Frequent Germline and Somatic Single Nucleotide Variants in the Promoter Region of the Ribosomal RNA Gene in Japanese Lung Adenocarcinoma Patients"

_cells, 2020, doi:10.3390/cells9112409_

Round 1

Reviewer 1 Report

The manuscript by Riuko Ohashi et al. identified the single nucleotide variants (SNVs) in the rDNA promoter region, in 82 lung adenocarcinomas they discovered that 29 tumours carried germline SNVs, and 8 tumours harboured somatic SNVs.

Although I am agreeing with the authors when they are saying that (line 68 to 73):

“While most cancer genome studies focused exclusively on variants that alter the amino acid sequences of protein-coding genes, in recent years, a new and crucial role of ncRNA in cancer has been highlighted. Although initially ncRNAs were thought to be "junk" with no functional purpose, trailblazing research has discovered their diverse functions in coding, decoding, transfer, regulation, gene expression, and complex integrated networks. Moreover, they have been identified as tumor suppressors and oncogenic drivers. Alterations in ncRNA expression and their mutations can promote cancer development and progression”

It is important to clearly mention in the manuscript that it is been recognized for a long time now, the huge contribution of ncRNAs in the cancer development, thinking for example at microRNAs, at long non-coding RNAs in cancer.

As the authors mentioned the rRNA alteration leads to defect on ribosome biogenesis that it is associated with a class of pathologies called “ribosomopathies”.

Here just a suggestion for the authors, but it would be interesting to know if the SNVs identified in the promoter region were also linked with a defect on ribosome biogenesis or with the induction of ribosomal stress (activation of p53 for example).

A possible experiment it could be to induce the ribosomal stress in cells, using compounds that causes defect on ribosome biogenesis (for example rapamycin, Camptothecin, actinomycin D low doses) and either to test the phosphorylation level of p53 or the level of transcription/processing of rRNA and comparing one human cancer cell lines identified to carry the SNVs (table1), with another which is not having this SNVs in the rDNA promoter region.

The founding, that the authors presented in this article, it is very interesting and very informative, the analysis the SNVs in the rDNA promoter region could lead to further future development, considering the possible effect on UBF binding with the promoter, that could have further regulation of rRNA transcription. In my opinion, a very important experiment, for the follow up of the project, would be to test via CHIP (chromatin immunoprecipitation) the affinity of UFB to the promoter in cells carrying the SNVs with cells without.

Minor points:

  • Line 339 is mention table 4 but there is not “table 4” included in the manuscript. The authors should correct the mistake
  • Table 2 is split into two pages which makes it hard to read it.
  • Lines 58 and 59, an additional reference is needed to justify the sentence that “Lung cancer is the leading cause of cancer-related deaths worldwide, and non-small cell lung cancer (NSCLC) accounts for most cases (approximately 85%).

Reviewer 2 Report

In this manuscript, the authors have studied mutation in the promoter region of rDNA genes in 82 lung adenocarcinoma.  Both germline and somatic mutations were found.  Interestingly, germline mutations within +1 to +100 from transcriptional start sites, were associated with shortened survival of lung cancer patients.  This is not the case for somatic mutations. The authors propose that germline mutations in rDNA promoter may have prognostic values.

Major critiques:

  1. The cohort sample size is not large enough, and there is no validation study to confirm this interesting finding. Therefore, the discovery of prognostic germline mutations may be simply caused by the unusual and particular composition of the patient cohort used under this study and cannot be generalized. In this sense, the result of this study is weak and not strong enough to support the conclusions. TCGA may have pan-cancer WGS data and can be used to test the authors’ hypothesis.
  2. The involvement of germline mutations rather than somatic mutations is perplexing. Are these germline mutations linked to lung cancer somehow, and semi-revealed by previous GWAS studies?
  3. The function of these germline mutations (around +50) in terms of affecting ribosomal biogenesis is not studied. Among the 20 cell lines tested, do those carrying such mutations show differential rRNA expression, compared to those without such mutations?

Round 2

Reviewer 2 Report

The revised manuscript did not properly address my concerns:

(1) The authors did not show interest to properly validate the finding using TCGA WGS data.

(2) There is no previous WGAS study results supporting their findings.

(3) New Figure 8 results are very confusing: how 45S pre-rRNA and 28S pre-rRNA (or is this mature 28S rRNA?) were detected; the difference in PC9 after actinomycin D treatment (45S) is apparent and cannot be statistically insignificant; the different responses between 45S and 28S need explanation. 
